

# $tW$ and $tZ'$ production at hadron colliders

**Nikolaos Kidonakis⋆, Marco Guzzi and Nodoka Yamanaka**

Kennesaw State University, Kennesaw, GA 30144, USA

⋆ nkidonak@kennesaw.edu

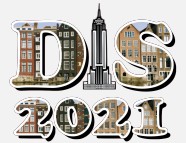

*Proceedings for the XXVIII International Workshop
on Deep-Inelastic Scattering and Related Subjects,
Stony Brook University, New York, USA, 12-16 April 2021*

## Abstract

We present theoretical results with soft-gluon corrections for two separate processes: (1) the production of a single top quark in association with a $W$ boson in the Standard Model; and (2) the production of a single top quark in association with a heavy $Z'$ boson in new physics models with or without anomalous couplings. We show that the higher-order corrections from soft-gluon emission are dominant for a wide range of collider energies. Results are shown for the total cross sections and top-quark transverse-momentum and rapidity distributions for $tW$ and $tZ'$ production at LHC and future collider energies up to 100 TeV. The uncertainties from scale dependence and parton distribution functions are also analyzed.

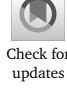

## 1 Introduction

Processes involving the associated production of a top quark with electroweak bosons in the Standard Model and beyond are very interesting and useful in determinations of various parameters and in the exploration of new physics. Thus it is imperative to have good theoretical predictions for the cross sections of these processes at hadron colliders. Perturbative QCD corrections are typically dominated by soft-gluon contributions which are usually large and, thus, important in such processes.

In particular, for the associated production of a top quark with a $W$ boson, it is known that QCD corrections at higher orders are considerable, and the soft-gluon corrections [1–4] are numerically dominant even at very high collider energies that are far from threshold [5]. The associated production of a top quark with a $Z'$ boson [6] in various models with new physics, with or without anomalous top-quark couplings, is also dominated by soft-gluon corrections.

The soft-gluon corrections appear in the perturbative expansion as logarithms of a kinematical variable, $s_4$, that vanishes at partonic threshold. We take Laplace transforms of the

partonic cross section as $\hat{\sigma}(N) = \int (ds_4/s) \, e^{-Ns_4/s} \hat{\sigma}(s_4)$, with transform variable $N$. The factorized expression for the cross section for $tX$ production (where $X$ stands for either a $W$ or a $Z'$ boson) is

$$\sigma^{f_1 f_2 \to tX}(N) = H^{f_1 f_2 \to tX} \, S^{f_1 f_2 \to tX} \left( \frac{m_t}{N \mu_F} \right) \psi_1 (N_1, \mu_F) \, \psi_2 (N_2, \mu_F) \,, \tag{1}$$

where $H^{f_1 f_2 \to tX}$ is an $N$-independent hard function, $S^{f_1 f_2 \to tX}$ is a soft function that describes noncollinear soft-gluon emission [1, 2], $\psi_1$ and $\psi_2$ describe collinear emission from initial-state partons $f_1$ and $f_2$ [7], $m_t$ is the top-quark mass, and $\mu_F$ is the factorization scale.

The soft function $S^{f_1 f_2 \to tX}$ obeys a renormalization group equation in terms of a soft anomalous dimension $\Gamma_S^{f_1 f_2 \to tX}$ [1, 2, 4] that controls the evolution of the soft function and which, together with the evolution of the $\psi$ functions [7], gives the exponentiation of logarithms of $N$ in the resummed cross section. The resummed expression is then expanded at finite order so as to provide predictions for physical cross sections upon inversion to momentum space.

The approximate NLO (aNLO) results, i.e. the LO cross section plus the NLO soft-gluon corrections, are very good approximations to the complete NLO results for the $tW$ and $tZ'$ processes. When we add the NNLO soft-gluon corrections to the complete NLO results, we obtain approximate NNLO (aNNLO). The further addition of $\mathrm{N}^3\mathrm{LO}$ soft-gluon corrections provides approximate $\mathrm{N}^3\mathrm{LO}$ (a$\mathrm{N}^3\mathrm{LO}$) results.

## 2 $tW$ production

We begin with results for $tW$ production as were most recently discussed in Ref. [5].

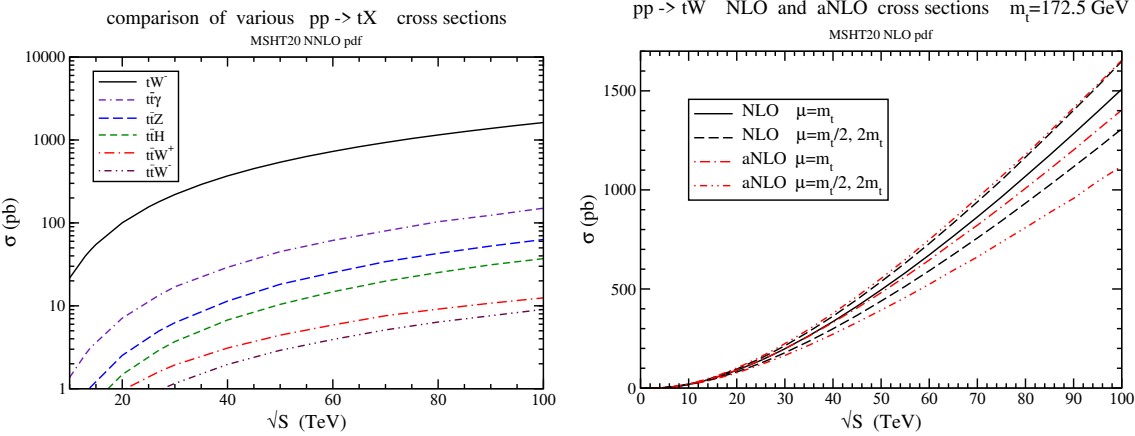

Figure 1: (left plot) A comparison of various $tX$ cross sections; (right plot) NLO and aNLO cross sections for $tW$ production.

We first note that $tW$ production cross sections are larger than for other top-quark processes with final-state electroweak and Higgs bosons. The left plot in Figure 1 shows for comparison the cross sections for a variety of processes at collider energies where it is seen that $tW$ cross sections are orders of magnitude larger than other ones over a large energy range. In this and the other plots in this section we have used MSHT20 parton distribution functions (pdf) [8].

Concentrating on the $tW$ process, we find that, remarkably, the aNLO cross section is a very good approximation to the complete NLO result for all forseeable collider energies, through 100 TeV, which shows that the soft-gluon corrections are dominant. The plot on the right in Figure 1 shows that the aNLO and NLO results, including scale variation, are very close to each

other. We have used MADGRAPH5_AMC@NLO [9] for the complete NLO results with removal of diagrams with resonant $\bar{t}$ contributions to avoid overlap with $t\bar{t}$ production (this does not affect the soft-gluon contributions, see discussion in Ref. [3] and references therein).

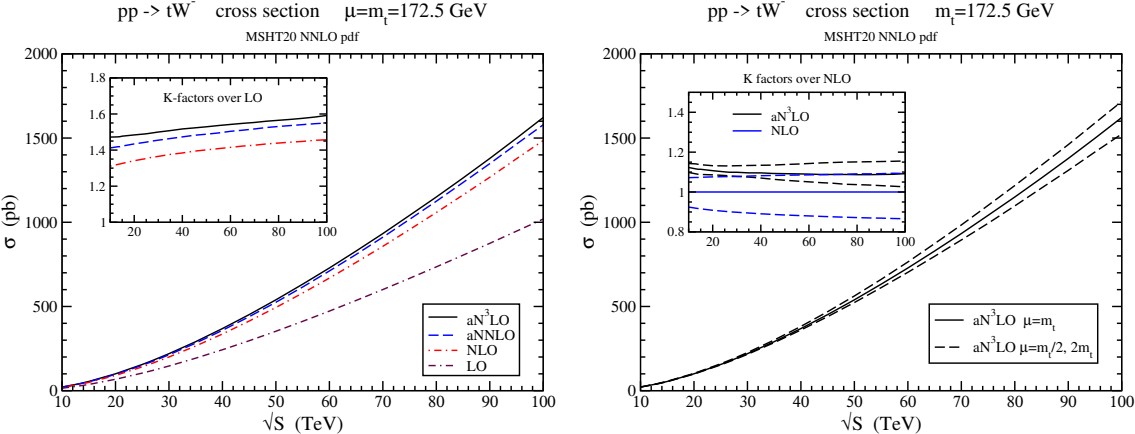

Figure 2: Cross sections for $tW$ production at collider energies.

The aNNLO and aN$^3$LO corrections (derived from resummation at NNLL accuracy) are also significant. On the left plot of Figure 2 we display the cross sections for $tW$ production at LO, NLO, aNNLO, and aN$^3$LO. The inset shows the $K$-factors relative to LO. The plot on the right shows the scale variation of the aN$^3$LO result, and the inset compares that to the variation at NLO; as expected, the aN$^3$LO scale dependence is significantly reduced relative to NLO.

All these theoretical predictions are in very good agreement with LHC data at 7, 8, and 13 TeV energies [10–12]. The aN$^3$LO cross section for the sum of the $tW^-$ and $\bar{t}W^+$ processes at 13 TeV is $79.5^{+1.9+2.0}_{-1.8-1.4}$ pb, where the first uncertainty is from scale variation from $m_t/2$ to $2m_t$, and the second uncertainty is from the MSHT20 pdf.

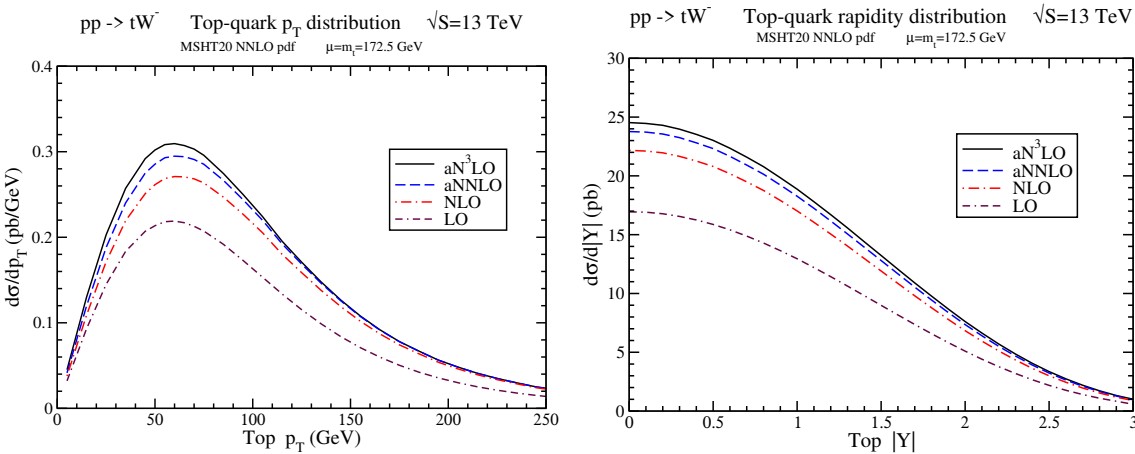

Figure 3: Top-quark $p_T$ (left) and rapidity (right) distributions in $tW$ production.

The top-quark $p_T$ and rapidity distributions in $tW$ production at 13 TeV energy are displayed in Figure 3. Again, we observe that the higher-order corrections are large in both distributions. The $W$-boson distributions may also be found in Ref. [5].

# 3 $tZ'$ production

We continue with results for $tZ'$ production in two different new-physics models [6]. The first model involves anomalous $t$-$u$-$Z'$ or $t$-$c$-$Z'$ couplings that contribute flavor-changing neutral-current (FCNC) terms in the Lagrangian, $\mathcal{L}_{FCNC} = (\kappa_{tqZ'}/\Lambda) e\, \bar{t}\, \sigma_{\mu\nu} q\, F_{Z'}^{\mu\nu} + $ h.c. [6]. The second model involves initial-state top quarks with interactions $\sum_{i=L,R} z_{t,i} g_{Z'} \bar{t}_i \gamma^\mu t_i Z'_\mu$ [6, 13, 14].

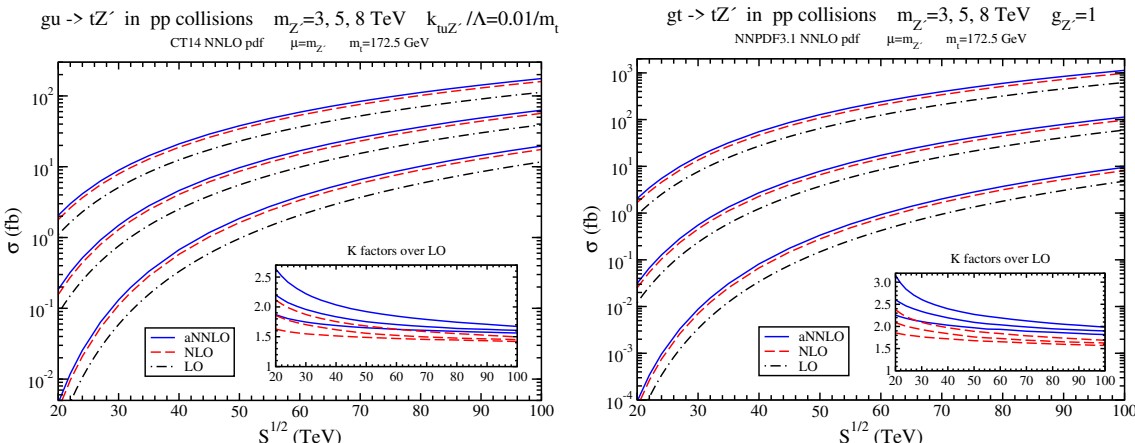

Figure 4: Cross sections vs. collider energy for $tZ'$ production via anomalous $t$-$u$-$Z'$ couplings (left) and via initial-state tops (right).

In Figure 4 we show LO, NLO, and aNNLO cross sections for $tZ'$ production via anomalous $t$-$u$-$Z'$ couplings using CT14 pdf [15] in the plot on the left, and via initial-state tops using NNPDF3.1 pdf [16] on the right. The cross sections are shown for a large range of collider energies up to 100 TeV for three different choice of $Z'$ mass. The $K$-factors relative to LO are shown in the respective inset plots.

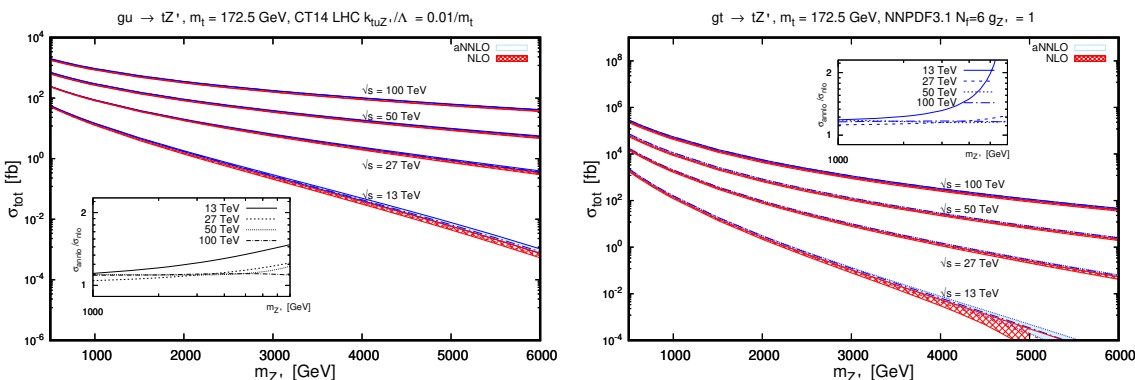

Figure 5: Cross sections vs. $Z'$ mass for $tZ'$ production via anomalous $t$-$u$-$Z'$ couplings (left) and via initial-state tops (right).

In Figure 5 we show the $tZ'$ cross sections, including pdf uncertainties, via anomalous $t$-$u$-$Z'$ couplings on the left and via initial-state tops on the right. The cross sections are shown for a large range of $Z'$ masses up to 6000 GeV for four different choices of collider energy. $K$-factors relative to NLO are shown in the insets.

We finally note that the top-quark $p_T$ and rapidity distributions in $tZ'$ production in both models of new physics also receive large corrections at higher orders [6].

## 4 Conclusion

We have presented results for $tW$ and $tZ'$ production in hadron colliders. We have demonstrated in all cases that higher-order corrections are large and are dominated by soft-gluon contributions. The inclusion of these corrections at higher orders improves the theoretical predictions for total cross sections and differential distributions.

**Funding information** This material is based upon work supported by the National Science Foundation under Grant No. PHY 1820795 (for N.K. and N.Y.) and under Grant No. PHY 1820818 (for M.G.).

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
