# Peer review of "$tW$ and $tZ'$ production at hadron colliders"

_SciPost Physics Proceedings, doi:SciPost Phys. Proc. 8, 064 (2022)_

## Round 1 · Referee Report · Anonymous (Referee 2) · 2021-8-10

Report

The authors have addressed the issues in my previous report satisfactorily.

---

## Round 1 · Referee Report · Anonymous (Referee 1) · 2021-8-25

Report

I appreciate the fact that the treatment of ttbar overlap has been mentioned.
Still, I do consider that the importance attributed to soft-gluon corrections is exaggerated and spelled in a misleading way . This also reflects the comments by Referee 3.
I also stress that cases are known where predictions based on soft-gluon resummation fail to reproduce the exact higher-order results, see eg https://arxiv.org/pdf/1303.0693.pdf for some top-related studies.

This said I leave the final decision to the editor in charge.
  • validity: -
  • significance: -
  • originality: -
  • clarity: -
  • formatting: -
  • grammar: -

Author:  Nikolaos Kidonakis  on 2021-08-25  [id 1708]

(in reply to Report 2 on 2021-08-25)

This referee report on our resubmitted manuscript makes incorrect statements on the quality of the soft-gluon approximation that we already addressed in our response to referee 3 of our original submission. The referee points to an older conference proceedings that uses a different resummation method, which one of us (N.K.) has shown in previous papers to be problematic, and which indeed failed to predict the correct size of the higher-order corrections. Therefore, pointing to that work does not show that our statements are misleading; rather, it shows that the method used in that work is not adequate, as has been known for a long time. We refer the referee to the large literature (including several papers by N.K., e.g. Phys. Rev. D 64, 014009 (2001); arXiv:1311.0283; Phys. Rev. D 96, 034014 (2017) ) that shows the differences between various resummation methods and the relative successes (or failures) of those methods. And, of course, the results in our proceedings contribution (figure 1, right plot) speak for themselves.

Author:  Nikolaos Kidonakis  on 2021-08-27  [id 1712]

(in reply to Nikolaos Kidonakis on 2021-08-25 [id 1708])

We strongly agree that soft gluon resummation should always be benchmarked against fixed-order predictions at NLO and NNLO, when available, to test its quality, and that's exactly what we have been doing in this paper and in past work. But our point is that not all soft-gluon formalisms are equal, so the failure of one formalism does not have implications for another one. The formalism we have used has produced very good approximations for a multitude of processes in over two decades of work and publications. Again, we refer the referee to the literature.

The numerical discussion that the referee has given in a comment is not quite correct. At 100 TeV energy, the soft-gluon aNLO corrections are practically identical to the full NLO corrections (within around a percent) for mu=m_t/2, they are 80% of the full corrections for mu=m_t, and they are 65% (NOT half) of the full corrections for mu=2m_t. So they are always dominant, even in the most extreme case. For smaller energies, the approximations get increasingly better (the lines are on top of each other) but that has been known for over a decade, so the focus of our current work was on the remarkable fact that the approximation works very well even at very high energies.

Anonymous on 2021-08-26  [id 1710]

(in reply to Nikolaos Kidonakis on 2021-08-25 [id 1708])

In my previous comment, my point on soft gluon resummation was that it should always be benchmarked against fixed-order predictions, when available, before making any strong statement about its goodness, exactly for the reason that cases are known where such an approach fails.

About the quality of the agreement of figure 1, right plot. The only thing I can deduce is that the claimed agreement is scale-dependent, with discrepancies between the approximate and exact NLO prediction which can reach 20% at 100 TeV (1100 vs 1300 pb), half of the size of NLO corrections at that energy. At lower energies, drawing any conclusion from that plot is impossible (all curves are on top of each other, and the scale is very compressed), perhaps a logarithmic scale or a ratio NLO/aNLO could provide more information.

---

## Round 1 · List of Changes

We have added a reference to the NLO calculation and a brief description of how we have dealt with the removal of overlap with ttbar production.

---

## Editorial Decision

published